# WLiT: Windows and Linear Transformer for Video Action Recognition

**DOI:** 10.3390/s23031616

**Published:** 2023-02-02

**Authors:** Ruoxi Sun, Tianzhao Zhang, Yong Wan, Fuping Zhang, Jianming Wei

**Affiliations:** 1Shanghai Advanced Research Institute, Chinese Academy of Sciences, Shanghai 201210, China; 2School of Information Science and Technology, ShanghaiTech University, Shanghai 201210, China; 3School of Electronic, Electrical and Communication Engineering, University of Chinese Academy of Sciences, Beijing 100049, China; 4State Key Laboratory of Geomechanics and Geotechnical Engineering, Institute of Rock and Soil Mechanics, Chinese Academy of Sciences, Wuhan 430071, China

**Keywords:** action recognition, Spatial-Windows attention, linear attention, self-attention, transformer

## Abstract

The emergence of Transformer has led to the rapid development of video understanding, but it also brings the problem of high computational complexity. Previously, there were methods to divide the feature maps into windows along the spatiotemporal dimensions and then calculate the attention. There are also methods to perform down-sampling during attention computation to reduce the spatiotemporal resolution of features. Although the complexity is effectively reduced, there is still room for further optimization. Thus, we present the Windows and Linear Transformer (WLiT) for efficient video action recognition, by combining Spatial-Windows attention with Linear attention. We first divide the feature maps into multiple windows along the spatial dimensions and calculate the attention separately inside the windows. Therefore, our model further reduces the computational complexity compared with previous methods. However, the perceptual field of Spatial-Windows attention is small, and global spatiotemporal information cannot be obtained. To address this problem, we then calculate Linear attention along the channel dimension so that the model can capture complete spatiotemporal information. Our method achieves better recognition accuracy with less computational complexity through this mechanism. We conduct extensive experiments on four public datasets, namely Something-Something V2 (SSV2), Kinetics400 (K400), UCF101, and HMDB51. On the SSV2 dataset, our method reduces the computational complexity by 28% and improves the recognition accuracy by 1.6% compared to the State-Of-The-Art (SOTA) method. On the K400 and two other datasets, our method achieves SOTA-level accuracy while reducing the complexity by about 49%.

## 1. Introduction

One of the most important tasks in video understanding is to understand human actions. The task to recognize human actions in a video is called video action recognition [1]. Transformer architecture is driving a new paradigm shift in computer vision, and researchers are rapidly adapting transformer architectures to improve the accuracy and efficiency of action recognition task [2]. Video frame sequence has an explicit sequential relationship-like sentence, which means that video understanding has a high similarity to Natural Language Processing (NLP) tasks [3]. Therefore, rich context information is essential for video understanding tasks. This is one of the reasons that the Transformer structure [4], which is widely used in NLP, has received a lot of attention in the video field in recent years. A unit in convolutional networks only depends on a region of the input, and this region in the input is the perceptual field for that unit. Since anywhere in an input image outside the perceptual field of a unit does not affect the value of that unit, it is necessary to carefully control the perceptual field to ensure that it covers the entire relevant image region [5]. So, another important reason for the widespread use of the Transformer structure is that the size of the perceptual field is important for many Computer Vision (CV) tasks, especially for video understanding tasks. The main previous approach is a three-dimensional convolutional neural network [6,7,8,9] (3D CNN) using 3×3×3 convolutional kernels with multilevel architectures and down-sampling to gradually increase the perceptual field. However, when the Transformer [10,11] has been widely used in the CV field in the last two years, it becomes a better approach to capture long-distance visual dependencies through the attention layer. While this approach has a powerful global context modeling capability, its computational complexity grows quadratically with token length, limiting its ability to scale to high-resolution scenarios. Therefore, designing a more efficient model structure that can capture global information is still an open issue. There is much effort that has been undertaken to modify the Transformer’s structure to pursue this goal.

IGPT [10] uses the standard Transformer model to solve vision tasks with the pixels of the input image as the sequence to be processed, and the information interacts at the pixel level. Subsequently, ViT [11] divides the image into non-overlapping patches instead of pixels and imitates the name in the NLP field [12], calling each block a “token”. That effectively reduces computational effort and improves performance and lays the foundation for later research. To further reduce the computational cost, several approaches propose to compute local attention within a window [13,14,15,16], as well as to perform spatial down-sampling operations on tokens [17,18] during attention computation. Such methods maintain good accuracy while significantly reducing the computational effort, but they still require operations such as shifting [13], overlapping patches [17,18], etc., to compensate for the lost spatiotemporal information. In addition to this, some methods note that the attentional overhead is mainly in the intermediate matrix operations [19,20,21]. In particular, the Softmax function restricts the order of matrix computation, resulting in a complexity that is a squared relationship of the sequence length. They try some operations to change the computational order of the matrix, but the results are not satisfactory. These methods obtain a trade-off between accuracy and computational complexity, but it is clear that researchers still pursue better methods that reduce complexity while maintaining the most complete spatiotemporal information possible.

This paper aims to propose a more accurate and efficient action recognition method, which can reduce the computational complexity as much as possible while ensuring recognition accuracy. Our work is inspired by the above works. Firstly, we divide the feature maps into multiple windows along the spatial dimensions and calculate the attention inside the windows, respectively. Unlike Swin [22], which divides the windows along the spatiotemporal dimensions, we only divide the windows along the spatial dimensions, which further reduces the computational complexity. This method also brings the problem of a limited perceptual field of the model; that is, the model can only represent and reinforce the information inside each window and lacks the ability to mine long-distance dependencies. Therefore, we consider that features extracted from different dimensions may help to solve this problem. In traditional attention calculations [16], each feature in the spatiotemporal dimensions is called a spatiotemporal token, which has complete channel information. Thus, if we transpose the dimensions of the feature maps, that is, each feature in the channel dimension is called a channel token, which contains complete spatiotemporal information. As shown in Figure 1, we divide the feature maps along the spatial and channel dimensions to obtain the Spatial-Windows token and Channel token, respectively. After the Spatial-Windows attention calculation, we use the Channel token for attention calculation, which we call Linear attention, so as to supplement the model’s ability to represent global information. Different from some of the previous Linear attention methods, we still follow the self-attention without introducing other additional operations to the calculation process. To intuitively represent the logical positional relationship between Spatial-Windows attention and Linear attention, other modules between the two attentions are omitted.

While such a method is theoretically feasible, there are still some challenges to overcome. In the calculation of attention, the Spatial-Windows/Channel token takes the channel/spatiotemporal dimensions as the hidden dimensions, and the complete channel/spatiotemporal information of a Spatial-Windows/Channel token is compressed into a single value after matrix multiplication during the calculation. This means that channel/spatiotemporal information is naturally lost in complex operations. Therefore, as shown in Figure 1, we use two kinds of attention alternately to compensate for the lost information in the spatiotemporal and channel dimensions.

To summarize, the main contributions of this article are as follows:(1)We propose a complementary framework of Windows and Linear Transformer (WLiT), which ensures the ability of the model to capture global information while achieving efficient action recognition.(2)We present the Spatial-Windows attention module that only divides the feature maps along the spatial dimensions, which further reduces the computational complexity.(3)We fully analyze and discuss the computational complexity of the attention mechanism, and theoretically prove our method.(4)We conduct a lot of experiments to verify our method. On the SSV2 dataset, our method achieves higher accuracy than the SOTA method while having less computational complexity.

This paper consists of five parts: The first part is the introduction, which introduces the action recognition task and the mainstream methods in this field and summarizes the methodological basis and main contributions of our study. The second part introduces some work related to our research and describes the problems and optimization possibilities of the previous methods in detail. The third part describes our research method particularly, not only introducing the overall structure of the method but also fully interpreting each part of the model. The fourth part first introduces the dataset selected in this paper and the relevant experimental details and then shows sufficient experimental results to prove the reliability of our research. The last part summarizes the content of the full text and puts forward possible directions for further research.

## 2. Related Works

The mainstream methods of action recognition can be divided into two categories according to the order of development, namely convolution-based methods and Transformer-based methods. To describe our work more clearly, we will also briefly describe the basic Transformer structure and self-attention in the second part.

### 2.1. Convolution-Based Action Recognition Methods

The earliest models relied on manually extracted features [23,24,25] to encode motion and scene information. With the advent of large datasets such as ImageNet [26], Convolutional Neural Networks [27] are widely used for various vision tasks and are also rapidly becoming the backbone of video understanding tasks [28,29,30]. Since the release of the Kinetics dataset [8], 3D CNN [7,31] has been widely used, and many variants [32,33,34,35] have emerged to improve the accuracy and efficiency of convolutional models. However, 3D convolutional kernel is difficult to train, and it takes a lot of calculation costs. I3D [8] inflates the pre-trained 2D convolutional kernels for better optimization; it adapts mature image classification architectures to use for 3D CNN. Hence, I3D bypasses the dilemma that 3D CNNs have to be trained from scratch. Moreover, I3D ends the era where different methods report numbers on small-sized datasets such as UCF101 and HMDB51. Publications following I3D need to report their performance on Kinetics400 or other large-scale benchmark datasets, which push video action recognition to the next level [1]. Meanwhile, some works represented by P3D [36] and R2 + 1D [34] try to decompose the 3D convolution kernel to reduce the complexity. To be specific, a 3D kernel (e.g., 3×3×3) can be factorized to two separate operations, a 2D spatial convolution (e.g., 1×3×3) and a 1D temporal convolution (e.g., 3×1×1). P3D proposes three connection structures according to the order of spatial convolution and temporal convolution operations and how they affect the output results. In addition to this, some works [37,38,39,40] optimize the processing of convolution through different entry points to fully exploit the spatiotemporal modeling capability of convolution. TSM [38] shifts part of the channels along the temporal dimension, thus facilitating information exchanged among neighboring frames. In order to keep spatial feature learning capacity, they put temporal shift module inside the residual branch in a residual block. TSM has high industrial application value, but it also has some defects. TSM does not choose the shifted channels, so there will be some information confusion. TEA [39] uses motion features to recalibrate the spatiotemporal features to enhance the motion pattern. Although so many methods have been tried differently, there are still unavoidable problems with convolution-based methods. Since the convolution operations can only process one local area at a time, the perception field is limited. In the face of tasks sensitive to long-distance information, the performance of convolution methods is not satisfactory.

### 2.2. Transformer-Based Action Recognition Methods

The Transformer structure was first proposed for NLP tasks in 2017; it can accept all feature information input in parallel. The self-attention mechanism is an integral component of transformers, which explicitly models the interactions between all entities of a sequence for structured prediction tasks [41]. Therefore, transformer models have emerged as attractive and promising solutions very soon for improving the accuracy of challenging CV tasks such as action recognition [2]. For a given entity in a sequence of video frames, self-attention computes the dot-product of the query with all keys, i.e., the patches segmented from the video frames. The dot-product result is then normalized using a Softmax operator to obtain the attention scores. Each entity then becomes a weighted sum of all entities in the sequence, where the weights are given by the attention score [41]. The most prominent advantage of this structure is that it can process all input features at the same time, that is, it has a global perceptual field.

Video-related tasks require longer-distance information due to the additional temporal dimension. Therefore, Transformer blocks [4] are inserted into CNNs as additional layers to improve long-range interaction modeling of spatiotemporal features [42]. Over time, pure transformer architecture-based action recognition has started emerging with an increasing trend [2]. So based on the ViT [11,43], there are some methods [3,43,44,45] that propose different variants of the spatiotemporal information learning process, which not only validate the Transformer’s ability to capture spatiotemporal information over long distance but also enhance the ability of feature extraction and analysis. VTN [44] successfully modifies the Transformer structure in the image field to be a video recognition network. VTN solved the problem of how to transform the video sequence into the input form of Transformer and became a new baseline in the field. Subsequently, TimeSFormer [3] considers the differences between video and image and proposes five video-based attention schemes, including joint space-time attention (ST), divided space-time attention (T + S), and so on. In addition, it also successfully applies the residual connection in the CNNs to the video Transformer structure, which further improved the recognition accuracy. Based on these, ViViT [45] slices the feature maps along the spatiotemporal dimensions to obtain the 3D tokens. It completed action recognition with a pure Transformer structure and proposes two token construction methods and three temporal attention and spatial attention connection modes.

Although the Transformer structure shows superior competitiveness in extracting spatiotemporal long-range dependencies, it comes with a much larger computational and parametric volume than convolution. Therefore, several works attempt to design them in a lightweight manner. Swin [13] slices the feature maps into windows and calculates the local attention inside each window. This approach successfully reduces the sequence length by a factor of several, enhancing the local induction bias but also losing global modeling capability. For global information interaction, they use sliding windows with overlap to achieve this. Immediately afterward, Video Swin Transformer [22] was also proposed, which was used for video action recognition. They divided the video frame into 3D windows and performed the same operation. Video Swin Transformer achieved excellent recognition accuracy. MViT [46] uses a hierarchical structure and pool self-attention. The combination of the multi-scale idea and Transformer structure improves the accuracy while reducing the model complexity. However, the method has a large number of redundant operations for processing information, and there is still room for improvement. Therefore, we further explore the structure of ViT and try to propose a more concise and efficient action recognition model.

## 3. Method

Our method has been further innovated and optimized on the basis of the previous ones. We achieve the low computational complexity through Spatial-Windows attention and ensure the ability of the model to obtain global information through Linear attention and additional Feed-Forward-Network (FFN) modules. In addition, we use a concise adaptive position encoding module, which simply and efficiently ensures that the position of the tokens in spatiotemporal and channel dimensions is fixed. In this section, we first give an overview of the overall structure of the model and analyze the traditional spatiotemporal self-attention. Next, we detail the key components of the model one by one. Finally, we qualitatively analyze the computational complexity to prove the reliability of the proposed method and explain some key parameters in the model.

### 3.1. Overview of WLiT Architecture

We follow the structural setting of MViT [46] and other transformer-based methods [22] to facilitate a fairer comparison of results. As shown in Table 1, our WLiT is composed of 4 stages, each having several transformer blocks of consistent channel dimension. At the beginning of the network, we sample and crop the video to obtain input features of size 8×224×224 (8 is the number of frames, and 224 is the spatial resolution). WLiT initially projects the input to a channel dimension of D=64 with overlapping spatiotemporal cubes of shape 3×4×4. The resulting sequence of length 4×56×56=12,544 is reduced by a factor of 4 for each additional stage to a final sequence length 4×7×7=196 at the last stage. In tandem, the channel dimension is up-sampled by a factor of 2 at each stage, increasing to 512 at stage 4. To visually show the difference among stages, we also show the attention operator used in each stage and the number of superimpositions.

Figure 2 illustrates the architecture of our WLiT; it can be seen that each block contains an attention module, two adaptive position encoding modules, and an FFN module. The attention module can be divided into Spatial-Windows attention, Linear attention, and spatiotemporal self-attention at different stages. Spatial-Windows attention and Linear attention are the core of this study and are therefore presented in more detail in Figure 1 (The Norm layer, the adaptive position encoding layer, and FFN module in Figure 2 are omitted). A patch embedding layer is inserted before the start of each stage, and the adaptive position encoding module is used. The spatiotemporal resolution and feature dimensions are kept constant in each stage. What’s more, the FFN module is an important part of the transformer structure, which can introduce nonlinear feature activation and supplement the ability of the model to capture all channel information. Therefore, after each calculation of attention, the feature must be activated by an FFN module.

It should be emphasized that the attention modules in the first two stages of the model are different from those in the last two stages. We decide to use Spatial-Windows attention and Linear attention in the first two stages of the network. In the last two stages, we still use the traditional spatiotemporal self-attention. This is a good trade-off between computational complexity and precision that we achieve after theoretical analysis and experimental verification. We will conduct a sufficient analysis in the part of this section behind.

We first introduce the calculation process of spatiotemporal self-attention [3]. Assume that there is a video feature, x∈ℝN×C, where N=T×H×W denotes the length of the feature sequence; C denotes the channel dimension; T, H, W means time, height, and width, respectively. The feature x is projected by an adaptive matrix to generate QKV. Then we perform the calculation as follows.
(1)AQ,K,V =SoftmaxQKTdkV.

After the matrix multiplication in Q×K, the feature’s dimensions change from ℝN×C to ℝN×N. Besides, dk means the number of channels of K. The obtained similarity matrix is normalized by the Softmax operation and multiplied by V so that the feature’s dimensions change from ℝN×N to ℝN×C. Then, the features are output from the FFN module. In this process, the computational complexity is O(2N2C+12C2), which is mainly positively related to the sequence length as well as the number of channels.

In the first two stages of the model, N≫C, so we strive to minish N to reduce computational complexity. We achieve this through Spatial-Windows attention. Then, we supplement the global information with Linear attention. The calculation processes of Spatial-Windows attention and Linear attention are different from traditional self-attention, and we will introduce them in turn.

### 3.2. Spatial-Windows Attention

In the first two stages, the influence of sequence length on computational complexity dominates [47], so we introduce Spatial-Windows attention to limit the attention calculation to local windows. The whole feature maps are divided evenly in a non-overlapping manner. Suppose there are Nw windows, then each window contains n patches, where Nw=H×W×Tn, n=ws×ws, and ws is the window size. Additionally note that we divide the feature maps into windows only along the spatial dimensions, which means that our windows have only two dimensions, H and W. That is unlike the division method of Swin [22], and therefore each window contains fewer patches and is less computationally intensive.
(2)Aspatiotemporal (Q, K, V)= Awindowsq, k, v Nw,
where q, k, v∈ℝn×C stands for query, key, and value inside each window. The computational complexity of the process is O(2Nwn2C+12C2). It can be observed that there is a significant reduction in computational effort. The model loses the ability to mine global contextual information which can be captured by channel-based Linear attention. 

### 3.3. Channel-Based Linear Attention

Spatial-Windows attention is computed in a single window that does not overlap. It means that there is no interaction of information among different windows, which can severely limit the ability of the model to extract and represent features. We note that the Spatial-Windows token used in the calculation of Spatial-Windows attention contains all the channel information. So, if the feature maps are divided along the channel dimension, the obtained Channel tokens can cover all spatiotemporal information. Using Channel tokens for attention computation, we can effectively implement the interaction of different windows information. This improves the ability of the model to represent global information while maintaining a lightweight network. Therefore, after computing Spatial-Windows attention, we restore all windows to the original feature maps and then compute Linear attention on the entire feature maps.
(3)Achannel=SoftmaxKTVdkQ,
where Q, K, V∈ℝN×C represent the query, key, and value of Linear attention, respectively. It can be noticed that, unlike Spatial-Windows attention, the channels occupy the main component of the computational complexity in this process. We still follow the computational process of spatiotemporal self-attention without too much additional modification. This also proves that the effectiveness is derived from this structural design.

### 3.4. Adaptive Position Encoding

Compared with image tasks, video has an additional time dimension, which requires a more sufficient positional encoding layer to exploit the positional relationship of tokens. Instead of the usual absolute and relative position encodings [48,49], we choose to use a position encoding that adapts to different input lengths and does not require additional clipping or interpolation. The local inductive bias contained in the convolution method can naturally memorize the image position due to its unique characteristics such as translation invariance [50]. Some previous works used convolutions as a position encoding method [51]. We also follow such an idea but explore it a little deeper. At the beginning of each stage, we insert an adaptive position encoding to ensure that the positions among tokens are stable. Our method includes an operation of dividing the window. To ensure that this process does not affect the positional relationship of tokens, we additionally add an adaptive position encoding module that does not share weights before the features are input into the FFN module. In order to ensure the simplicity of the model, we choose depth-wise convolution to realize adaptive position encoding. We also verify in the experiments that the adaptive position encoding improves the accuracy of the model without increasing the number of parameters and computation.

### 3.5. Computational Complexity Analysis and Model Structure Design

We first analyze traditional spatiotemporal self-attention. The module of projecting from the feature maps to generate QKV is a linear layer, and the computational complexity is only related to the number of channels, that is, $O{(3C}^{2}).$ Then in the calculation process of attention, the complexity of QK multiplication is $O{(N}^{2}C)$. The complexity of multiplying the weight matrix by V is $O{(N}^{2}C)$. The complexity of the final linear layer and FFN module is $O{(9C}^{2})$. So, the total computational complexity is $O{(2N}^{2}{C+12C}^{2})$.

However, when we calculate the attention after dividing the feature maps into windows, the computational complexity becomes $O{(2N}_{w}n^{2}{C+12C}^{2})$, where $N_{w}$ means the number of windows, and n means the square of the window size. To show the reduction of computational complexity more intuitively, we roughly conduct a quantitative analysis. Taking the first stage as an example, N=56×56, C=64, so the first part of the traditional spatiotemporal self-attention complexity is ${2N}^{2}{C=2\times (56\times 56)}^{2}\times 64=1,258,815,488$. The complexity of Spatial-Windows attention is only ${2N}_{w}n^{2}{C=2\times 64\times (7\times 7)}^{2}\times 64=19,668,992$. Thus, the computational complexity is reduced by a factor of 64. Then there is Linear attention computed along the channel dimension. The computational complexity of other steps is the same as that of spatiotemporal attention, but only the complexity of the matrix multiplication process of attention becomes $O{(2C}^{2}N)$. So, the total computational complexity of Linear attention is $O{(2C}^{2}{N+12C}^{2})$. At the first stage, ${2C}^{2}{N=2\times 64}^{2}\times \left(56\times 56\right)=25,690,112$. The sum of the computational complexity of the Spatial-Windows attention and Linear attention is about 45,359,104, and the computational complexity is reduced by 27.8 times.

In the first two stages where N≫C, we reduce the computational complexity from $O{(N}^{2})$ to $O{(C}^{2})$. In the last two stages of the network, the spatiotemporal resolution has been reduced to a relatively low level due to the layer-by-layer down-sampling operation, which means N≪C. If we continue with the previous setup, the complexity is higher than normal attention. Thus, we only use Spatial-Windows attention and Linear attention in the first two stages of the network.

We sample 8 frames from a video as input, and we first down-sample the spatiotemporal dimensions of the frames using non-overlapping 2×4×4 convolutions and the input’s dimensions change from T×H×W×3 to $\frac{T}{2}\times \frac{H}{4}\times \frac{W}{4}\times 64$. After each stage, the input’s spatiotemporal dimensions are further reduced to $\frac{T}{2}\times \frac{H}{8}\times \frac{W}{8},\;\frac{T}{2}\times \frac{H}{16}\times \frac{W}{16},\;\mathrm{and}\;\frac{T}{2}\times \frac{H}{32}\times \frac{W}{32}$ by a 1×2×2 convolutional layer. Besides, the input’s channel dimension is raised from 3 to 128, 320, and 512. We set the channel expansion rate of FFN and the number of layers in the network stages completely following the settings in MViT [46]. Thus, the performance fluctuation caused by hyperparameter changes can be reduced as much as possible, and the effectiveness of our method is proved. The specific network structure is shown in Table 1.

## 4. Experiments

### 4.1. Setup

#### 4.1.1. Datasets

We conduct extensive experiments on four datasets. Deep learning methods generally improve accuracy when the amount of training data increases [1], especially for Transformer methods. Kinetics400 [8] and Something-Something V2 [52] are the two most widely used large-scale datasets in this field, so it is fair to provide recognition results on these two datasets. When the convolution-based methods are popular, UCF101 [53] and HMDB51 [54] are the most popular datasets, so we also verify the recognition results of the model. The K400 dataset contains 400 human action categories, including about 240 K training videos and 20 K verification videos. Then, we further validate the performance of the model on the SSV2 dataset [52] that is more concerned with temporal information. SSV2 dataset requires strong temporal modeling ability because most activities cannot be inferred based on spatial features alone [1]. It contains approximately 168.9 K training videos as well as 25 K validation ones with 174 classes. In addition, we also fine-tune and test our model on the UCF101 [53] and HMDB51 [54] datasets in Top-1 recognition accuracy. The specific number of categories and samples for each dataset is shown in Table 2.

#### 4.1.2. Implementation Details

In the training process, we use the uniform sampling strategy [9] for the K400 dataset. That is, we sample a clip from the full-length video, and the input to the network are T frames with a temporal stride of τ; denoted as T×τ [46]. We use the same training strategy as MViT [46]. In addition, to validate the capability of our model, we do not pre-train it on the large-scale ImageNet dataset but do the full training directly on the K400 dataset. For the SSV2 dataset, we load the weights pre-trained on the K400 and then train the model for 50 epochs. For the UCF101 and HMDB51 datasets, we also load the weights pre-trained on the K400 and fine-tune the model for 20 epochs. For all datasets, we use the AdamW optimizer [55] and set 10% of epochs as warmup. We extract eight frames as one clip on the K400 dataset, with an interval of eight frames, for a total of four clips from one video file, without spatial crop operations. We extract 16 frames at intervals of eight frames on the SSV2 dataset as a clip. A total of one clip and three spatial crop operations are used in the experiment. This is because the videos in the SSV2 dataset are more temporally related, so more frames are needed to ensure recognition accuracy. In our tests, we use the multi-clip test for the K400 dataset and the multi-crop test for the SSV2 dataset. For the UCF101 and HMDB51, we extract eight frames as one clip and take one clip to test. We train and test our model on four NVIDIA RTX 3090 GPUs.

### 4.2. Comparison with the State-Of-The-Arts

In this section, we compare the performance of our method with popular approaches in recent years on four datasets. For the fairness of comparison, we choose their small version as much as possible. The benchmark is the small version of MViT. In addition to Top-1 and Top-5 accuracy criteria, we also compare the number of parameters and calculations.

#### 4.2.1. Kinetics400

As shown in Table 3. We compare our approach not only with convolution-based methods but also with Transformer-based models that have performed prominently in recent years. There are some parts in the table from top to bottom, and we discuss them in turn. The first part shows the results of the convolution-based methods on the K400 dataset. Traditional convolutional methods naturally have a smaller number of parameters and calculations than Transformer-based methods, but they are also limited by the model capacity. It does not perform well when scaling up the model capacity and when facing, for example, K400 or even larger datasets. Our method has a larger perceptual field, which can better capture the global spatiotemporal information. Therefore, our method improves the accuracy by 0.5–3%. At the same time, our model has fewer layers than the convolutional methods, so the amount of calculation and parameters is less.

The second part is a variation of the various Transformer-based methods in recent years. Most Transformer-based methods use ImageNet for pre-training, which improves the accuracy by 2–3%. If the model is pre-trained on Imagenet21K, it can even be improved by about 5–7%. These models have a larger network size, more parameters, and higher computational complexity. The baseline criterion we choose, i.e., a small version of the MViT model without introducing additional pre-training parameters, is then shown. MViT performs down-sampling during the calculation of attention to reduce the spatiotemporal resolution of features, thereby reducing computational complexity. However, such a method will reduce the ability of features to represent whole information.

The last one is our WLiT model. Our method guarantees feature integrity as much as possible. We first use Spatial-Windows attention to extract features and then use Linear attention to supplement global information. Although the perceptual field of the model is limited by the window size, we have successfully further reduced the number of parameters and GFLOPs of the model. The best trade-off between accuracy and computational complexity has been achieved. It is easy to see that our method has 20% fewer parameters and 49% less computation compared to the benchmark, while the difference in accuracy is only 1.4% for Top-1 and 0.1% for Top-5.

#### 4.2.2. Something-Something V2

The performance of our method on the SSV2 dataset is shown in Table 4. This video dataset contains the behavior of object interactions and is referred to as a temporal modeling task. It pays less attention to textural information such as background and focuses more on temporal contextual information. The performance of the latest convolutional methods is shown first. Due to the limitation of its natural perceptual field, convolution-based methods cannot capture global spatiotemporal information well and perform relatively poorly on the SSV2 dataset.

The method based on Transformer has a global perceptual field, whose performance is more outstanding but also generates more parameters and computational effort. We choose the base version of MViT [46] as the baseline on this dataset. The SSV2 dataset is more sensitive to temporal information. The down-sampling operations of MViT on features significantly affect the model’s ability to represent temporal information, so the performance on this dataset is not satisfactory.

Although our method divides the feature maps along the spatial dimensions, which greatly reduces the computational complexity, it still ensures the integrity of the features. Subsequently, we strengthen the model’s ability to extract full temporal information by Linear attention. At the bottom of the table, we present the results of the test with 16 frames selected. Compared to the baseline, our method saves 28% computational complexity and 40% number of parameters while improving 1.6% and 2.3% in Top-1 and Top-5 accuracy, respectively. Even compared to X-ViT pre-trained on ImageNet21K, our method exceeds its 0.1% and 0.9% in Top-1 and Top-5 accuracy, respectively. In general, Table 4 verifies the capability of temporal modeling for WLiT, and our method achieves the best recognition accuracy with fairly less computational complexity on the SSV2.

#### 4.2.3. UCF101 and HMDB51

We compare the results of our method on the UCF101 and HMDB51 with other methods in Table 5. These two datasets that pay more attention to scene information are relatively small, suffering overfitting. Therefore, we pre-train the model on the K400 and then migrate the model to the UCF101 and HMDB51. Our method outperforms the SOTA method by 0.2% and 2.8% in Top-1 accuracy on the UCF101 and HMDB51 datasets, respectively.

There are two reasons behind this result. WLiT has strong robustness and can still achieve better recognition results in the face of low-resolution datasets. Different from other methods that pay more attention to appearance features, WLiT can also capture sufficient temporal information, so higher recognition accuracy is obtained.

### 4.3. Ablation Study

The SSV2 dataset requires a strong temporal modeling ability of the model [1]. We hope to prove that our method has the ability to capture global temporal information through sufficient ablation learning, so we choose to use the SSV2 dataset for ablation research. In addition, the scale of SSV2 datasets is relatively large, which can fully verify the performance of the algorithm. We show Top-1 accuracy and computational complexity in GFLOPs for a single clip and three spatial crop inputs of size 224×224.

#### 4.3.1. The Performance of the Linear Attention

Spatial-Windows attention limits the calculation scope to the inside of the window, obtains the information of the channel dimension, and reduces the computational complexity. However, the perceptual field of Spatial-Windows attention is small, and the model loses the ability to represent the global spatiotemporal information. Linear attention can extract global spatiotemporal information, which can help the model to make up for this deficiency. As shown in Table 6, when the Linear attention module is not used, the recognition accuracy is reduced by 4.7%. When the model can only focus on the information in the local window without global information interaction, the recognition accuracy of our method is greatly reduced. In addition, if the Spatial-Windows attention module is removed, the accuracy decreases by 4.1%. This is because Linear attention has limitations, which means that part of the spatiotemporal information will be lost in the calculation process of attention, so the recognition accuracy of the model also decreases significantly.

Besides, we also explore the effect of the combined order of the two attention modules on the model accuracy. As shown in Table 6, we show the results of stacking Spatial-Windows attention and Linear attention in different orders. It is obvious that computing the Spatial-Windows attention first and then computing the Linear attention can improve the recognition accuracy by 0.4% in Top-1. The model first extracts and enhances the information in local regions through Spatial-Windows attention and then mines the potential connections of each region through Linear attention. Therefore, it can obtain richer semantic information, which is also in line with biological studies.

#### 4.3.2. The Performance of the Extra FFN for the Linear Attention

The FFN module has always been a default part of the Transformer structure. So, we verify the impact of the FFN module on our WLiT. As shown in Table 7, the FFN module has little effect on the computational complexity, thus excluding the effect of network size on the recognition accuracy. The experimental results show that when the FFN in the Spatial-Windows attention module is removed, the Top-1 accuracy drops by 1%. When we delete the FFN in the Linear attention, the Top-1 accuracy drops by 1.2%. When all FFN modules in the network are not used, the accuracy decreases by 2.8% in Top-1.

The calculation process of attention is mainly matrix operations, all of which are linear operations. The FFN module introduces the nonlinear feature activation layer, which effectively helps the model to propagate gradients during the learning process. Besides, during the calculation of Linear attention, the part of channel information is lost. The FFN module can supplement the model’s representation ability for all channel information. Therefore, the FFN module is very important for WLiT, especially for the Linear attention module. We use the structure of alternating Spatial-Windows attention and Linear attention in each block, as shown in Figure 1, so unlike previous attention modules, we use the FFN module twice. Although the additional FFN module will introduce a small number of additional calculations and parameters, it also helps the model to improve the recognition accuracy.

#### 4.3.3. The Performance of the Adaptive Position Encoding

The position encoding module plays a crucial role in the network based on the Transformer structure, which ensures that the tokens can still perceive the relative positions of each other in the video frame when calculating the attention score. Therefore, we embed two adaptive position encoding modules in each block, as shown in Figure 2, respectively at the start of the attention calculation, and before the FFN module. This is because we make ensure that after dividing the feature maps into several windows and restoring them back, the tokens can still maintain the stable positional relationships. At the same time, to ensure the consistency of the model structure, we use this design in all attention modules. The results are shown in Table 8. We use depth-wise convolution to implement adaptive position encoding, so only a few parameters and computations are added. After deleting the first or second adaptive position encoding layer, the recognition accuracy of the model drops by 0.5% and 0.3%, respectively. When all adaptive position encoding layers are removed, the model accuracy drops by 2.2%. This can demonstrate the importance of adaptive position encoding.

#### 4.3.4. Empirical Investigation on Model Settings

We first analyze the effect of different window sizes on the model. Different window sizes correspond to different perceptual fields in the early stage of the model. The larger the window is, the less spatiotemporal information is split, but at the same time the number of computations increases. The results in Figure 3a show that when the window size exceeds seven, the accuracy only improves by 0.1%, but the computational complexity increases about 10%. Therefore, we find that setting the window size to seven is the best trade-off between computational complexity and accuracy.

Then, we modify part of the testing strategy to verify the robustness of our method. For the K400 dataset, we train the model with four clips which have eight frames in each, so we test the effect of different clips on the accuracy. As shown in Figure 3b, K400 is the key dataset for the scene, so multiple clips can focus on more scene information and improve performance. The experimental results show that when four clips are extracted for testing, the accuracy is close to the maximum. In other words, four clips can well summarize the content of the entire video, and the recognition ability of the model has reached saturation. If the number of clips continue to increase, it will only increase the computational complexity. Therefore, four clips are sufficient.

The SSV2 dataset is a temporal-related dataset and trained with a 16 frames clip, so multi-crop testing is better for capturing the motion information for boosting performance. The experiments in Figure 3c demonstrate that the number of clips has very little effect on test accuracy. This is because uniformly sampling one 16 frames clip can cover important information in the video. However, when the same number of clips is selected, the accuracy of the three crops is about 4.2% higher than that of the one crop. Multi-crop can more fully extract the object relationship on each frame, so multi-crop can obtain better recognition accuracy. However, increasing the number of crops will bring a significant increase in computational complexity, so we finally choose three crops for experimentation.

### 4.4. Visualization

To further demonstrate the effectiveness of our model, we visualize some outputs of our model. In Figure 4, we use Visualizer to show the attention heatmap output from the network. The red box in the graph means “query”, and the redder color in the graph means the higher attention score. According to the first graph of Figure 4a, it can be seen that the perceptual field of attention is limited to a small range using only the Spatial-Windows attention. Subsequently, the perceptual field is expanded into a larger region after Linear attention computation. Finally, the output from the last layer of the model shows that for the action of playing the piano, attention is focused on the keys when the “query” is the hand, which is a reasonable result.

In the visualization experiment, we extract 16 frames for prediction and then randomly select six frames for display, and the results are presented in Figure 4b. It is clear that the model effectively focuses on the important regions in all six displayed frames. This also confirms the effectiveness of our model.

## 5. Conclusions

We propose a pure Transformer architecture for action recognition based on two attention methods. We modify the traditional self-attention as the superposition of the Spatial-Windows attention and Linear attention and the directions of these two attention division tokens are different. Spatial-Windows attention divides the feature maps into Spatial-Windows tokens along the spatial dimensions, which contain full channel information with lower computational complexity. Linear attention divides the feature maps along the channel dimension to obtain Channel tokens, and these Channel tokens can cover all spatiotemporal information with long-range dependencies. We alternately use Spatial-Windows attention and Linear attention, which effectively improves the computational efficiency while ensuring powerful spatiotemporal information modeling. Our method is fully verified on four classical datasets as well as visualization experiments. We will further optimize our WLiT to achieve a more accurate recognition effect with less computational complexity. We will expand it to other video understanding tasks in the future.

## Figures and Tables

**Figure 1 sensors-23-01616-f001:**
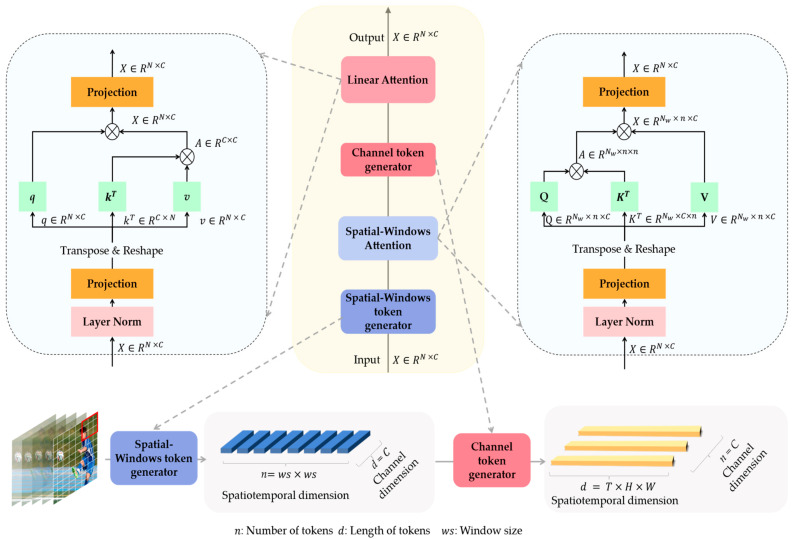
The logical pipeline of the Spatial-Windows Attention and Linear Attention with the generation of two kinds of tokens used in them. In addition, the attention calculation process and feature dimensions change during the pipeline are shown.

**Figure 2 sensors-23-01616-f002:**
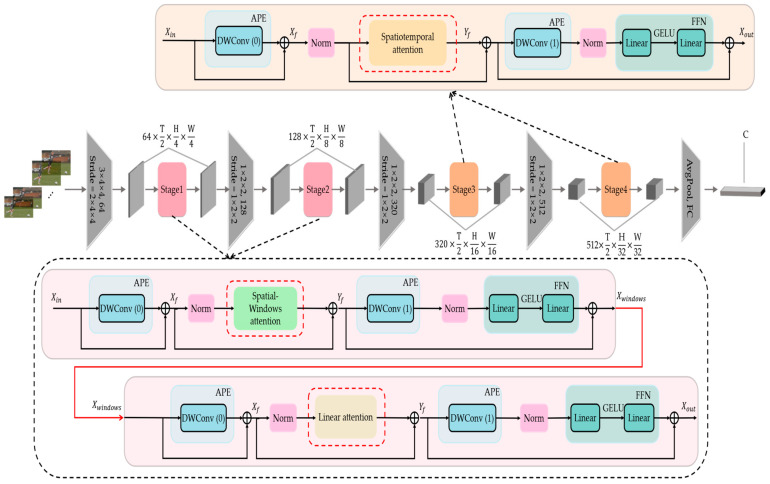
Overall architecture of our Windows and Linear Transformer (WLiT). We show one block of each stage of WLiT which contains three important parts: Adaptive Position Encoding (APE) to keep the position relationship between tokens stable, Feed-Forward-Network (FFN), which performs nonlinear activation on features, and the Attention Module. In the first two stages, we use Spatial-Windows attention and Linear attention. By alternately using the two types of attention, our model enjoys the benefit of capturing global information with lower computational complexity. In the last two stages, we use spatiotemporal self-attention.

**Figure 3 sensors-23-01616-f003:**
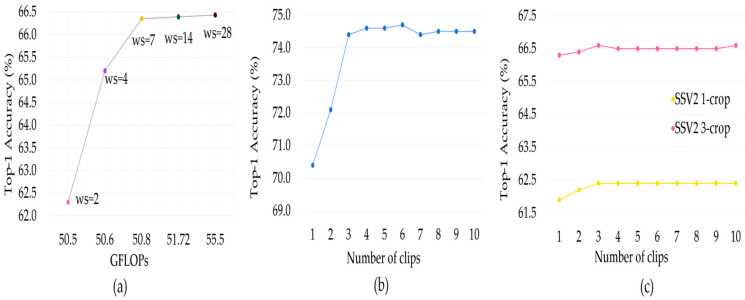
Experiments with some variables. (**a**) Effect of different window sizes on accuracy on the K400 dataset. (**b**) Results of accuracy on the K400 dataset with different number of clips. (**c**) Results of accuracy on SSV2 dataset with different number of clips and crops.

**Figure 4 sensors-23-01616-f004:**
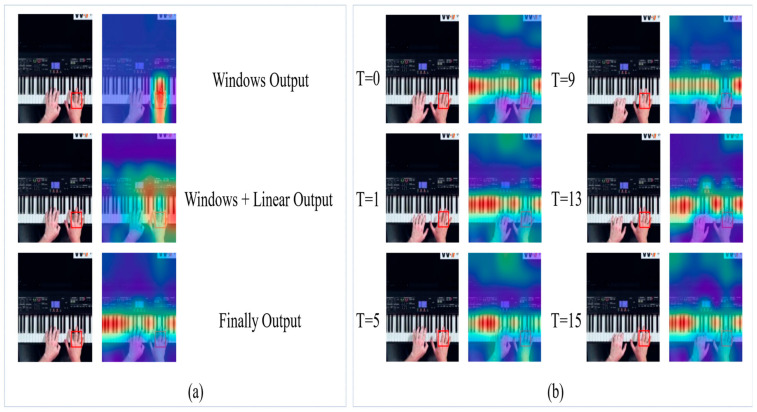
Some visualizations of WLiT. (**a**) We show the results of only Spatial-Windows attention calculation and then the results of Linear attention calculation and the final results. (**b**) We randomly select six of the sixteen frames we predict to show the attention heat-map.

**Table 1 sensors-23-01616-t001:** Model configurations for our WLiT.

Stage	Operators	Output Sizes
Pre-processing	Sampling 8×1×1	8×224×224
Patch embedding	Kernel 3×4×4, 64 Stride 2×4×4	64×4×56×56
Stage 1	$\left[\begin{array}{c} \text{Spatial-Windows}\;\text{attention}\\ \text{Linear}\;\text{attention} \end{array}\right]\times 1$	64×4×56×56
Stage 2	$\left[\begin{array}{c} \text{Spatial-Windows}\;\text{attention}\\ \text{Linear}\;\text{attention} \end{array}\right]\times 2$	128×4×28×28
Stage 3	$\left[\text{Spatiotemporal}\;\text{attention}\right]\times 11$	320×4×14×14
Stage 4	$\left[\text{Spatiotemporal}\;\text{attention}\right]\times 2$	512×4×7×7

**Table 2 sensors-23-01616-t002:** Public datasets of video action recognition.

Dataset	Category	Samples (Train)	Samples (Test)
Kinetics400 [8]	400	240,436	19,787
Something-Something V2 [52]	174	168,913	24,777
UCF101 [53]	101	9537	3734
HMDB51 [54]	51	3570	1530

**Table 3 sensors-23-01616-t003:** Comparison with previous work on the K400. We report the inference cost with a single “view” (spatial crop with temporal clip) multiplied by the number of views. GFLOPs (Giga floating point operations) for calculation amount and Mega for Parameters.

Method	Pre-Train	Frame	GFLOPs	Param.	Top-1 (%)	Top-5 (%)
Two-Stream I3D [8]	ImageNet	64	-	25.0	71.6	90.0
$R\left(2+1\right)$D [34]	-	32	75×1×10	61.8	72.0	90.0
bLVNet-TAM-24 × 2 [56]	Kinetics400	24	93.4×3×3	25.0	73.5	91.2
TSM [38]	ImageNet	8	33×1×10	24.3	74.1	91.2
STM [57]	ImageNet	16	-	-	73.7	91.6
ViT-B [46]	-	16	180×1×5	87.2	68.5	86.9
ViT-B-VTN [44]	ImageNet	250	4218×1×1	114.0	**78.6**	**93.7**
ViViT [45]	ImageNet21K	32	283.9×1×1	86.7	75.8	-
TimeSFormer [3]	ImageNet	8	590×3×1	121.4	75.8	-
MViT-S (Our baseline) [46]	-	8	32.9×1×5	26.1	76.0	92.1
WLiT (Ours)	-	8	** 20.9 × 1 × 4 **	**21.9**	74.6	92.0

**Table 4 sensors-23-01616-t004:** Comparison with previous work on the SSV2.

Method	Pre-Train	Frame	GFLOPs	Param.	Top-1 (%)	Top-5 (%)
bLVNet-TAM-32 × 2 [56]	-	32	128.6×3×3	40.2	65.2	90.3
MSNet-R50 [58]	-	16	67×1×1	24.6	64.7	89.4
Slow-Fast R101 [9]	K400	8	106×3×1	53.3	63.1	87.6
TSM [38]	K400	16	65×1×10	24.3	64.3	89.6
STM [57]	ImageNet	16	-	-	63.5	89.6
TEA [39]	ImageNet21K	16	70×3×10	-	65.1	89.9
TDN [40]	ImageNet	16	132×3×10	-	65.3	89.5
CTNet [59]	ImageNet	16	75×1×1	-	65.9	90.1
X-ViT [60]	ImageNet21K	32	850×3×1	-	66.2	90.6
ViViT-L [45]	K400	32	991×3×4	86.7	65.9	89.9
SSTSA-L [61]	ImageNet21K	32	1356×3×4	181.6	66.2	-
TimeSFormer [3]	ImageNet21K	16	5109×3×1	-	62.5	-
MViT-B (our baseline) [46]	K400	16	70.5×3×1	36.6	64.7	89.2
MViT-B [46]	K600	16	70.5×3×1	36.6	66.2	90.2
WLiT (Ours)	K400	16	** 50.7 × 3 × 1 **	**21.9**	**66.3**	**91.5**

**Table 5 sensors-23-01616-t005:** Comparison with previous methods on the UCF101 and HMDB51 in Top-1 accuracy.

Method	Pre-Train	UCF101 (Top-1%)	HMDB51 (Top-1%)
TSN [37]	ImageNet	94.0	68.5
P3D [36]	ImageNet	88.6	-
ARTNet [62]	K400	94.3	70.9
TSM [38]	K400	95.9	70.7
D3D [63]	K600	97.1	79.3
FASTER32 [64]	K400	96.9	75.7
Two-stream I3D [8]	K400	93.4	80.9
MEST [65]	ImageNet	96.8	73.4
WLiT (Ours)	K400	**97.3**	**83.7**

**Table 6 sensors-23-01616-t006:** Attention structure design.

Model	GFLOPs	Param.	Top-1 (%)	Top-5 (%)
Spatial-Windows attention	46.6	21.4	61.6	88.7
Linear attention	46.7	21.4	62.2	89.1
Linear → Windows	50.8	21.9	65.9	91.1
Windows → Linear (WLiT)	50.8	21.9	**66.3**	**91.5**

**Table 7 sensors-23-01616-t007:** Evaluation of the role of FFN in Spatial-Windows attention and Linear attention. ‘✗’ means deleting the FFN module.

FFN		SSV2			
In Spatial-Windows Attention	In Linear Attention	GFLOPs	Param.	Top-1 (%)	Top-5 (%)
✗	✗	45.8	21.3	63.5	90.1
✓	✗	48.3	21.6	65.1	90.5
✗	✓	48.3	21.6	65.3	90.7
✓	✓	50.8	21.9	**66.3**	**91.5**

**Table 8 sensors-23-01616-t008:** Measure the effect of two Adaptive Position Encoding (APE) modules.

APE		SSV2			
APE [0]	APE [1]	GFLOPs	Param.	Top-1 (%)	Top-5 (%)
✗	✗	50.5	21.7	64.1	90.2
✓	✗	50.7	21.8	66.0	91.3
✗	✓	50.7	21.8	65.8	91.4
✓	✓	50.8	21.9	**66.3**	**91.5**

## Data Availability

Not applicable.

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
