# Peer review of "WLiT: Windows and Linear Transformer for Video Action Recognition"

_sensors, 2023, doi:10.3390/s23031616_

Round 1

Reviewer 1 Report

The reviewed paper presented a model called windows and linear transformer (WLiT) for video action recognition. According to the authors, the combination of spatial-windows attention with Linear attention causes WLiT to be efficient for video action classification. In the last few years, transformers have been a good tool for solving image recognition tasks. Unfortunately, the reviewed manuscript is characterized by a lack of clear explanations and the null scientific rigor that is imperative in high-quality papers. There are many suggestions that authors should address before the possibility of accepting the paper for publication.

One of the main flaws is that authors use the terms recognition and classification as the same, which is wrong and confusing. How can the authors demonstrate that their proposal is efficient for video action classification/recognition?

Many issues regarding English language usage should be addressed. Moreover, please revise many typo errors. All the acronyms should be explained on the first usage. Please insert a space between the word and the call for reference in the entire document.

The abstract was well-written and explained. However, many asseverations were not supported by scientific evidence. For example, the authors commented that "transformer has led to the rapid development of video understanding." However, no reference was included to support this.

The authors should insert information regarding context in Section I. Please insert a reference for the first two sentences of the introduction. In addition, the authors should insert video understanding and action recognition information in the introduction. This information is essential to offer a context for the reader. Moreover, please state clearly the aim of the paper.

Many key terms should be explained so readers can understand the article better. For example, what is a token? What is a perceptual field? What is the attention mechanism? Please describe in-depth the information presented in Figure 1. At the end of the introduction, please insert a paragraph describing the paper's organization.

The authors should include a section to explain the transformer's theoretical background. This will help understand the paper better and highlight the proposed main differences.

At the beginning of section II, please include a paragraph explaining the section's organization. The information presented in the related work is vague. The analysis of each paper should be conducted in-depth. Building a table that can summarize the main features of the works analyzed should be desirable. Moreover, many interesting references were missed. Therefore, it should be desirable that authors conduct a better literature search. For example, please revise the following papers:

-          Vision Transformers for Action Recognition: A Survey.

-          Transformers in Vision: A Survey.

-          An effective video transformer with synchronized spatiotemporal and spatial self-attention for action recognition.

Undoubtedly, Section III regarding method is the poorest of the whole paper. Most of the explanations were not clear. Please avoid exaggerations. All the arguments presented should be supported by scientific evidence (references). Please describe in which sense the proposal is innovative. Please demonstrate that the proposal is optimum. How the model obtained the global information? Why was the model divided into four stages? Why each block contains an attention module, two adaptive position encoding modules, and an FFN module?

Please explain in-depth the information presented in Table 1. Please increase the quality of the figures. Figure 2 should be explained better. Labeling Figure 2 according to the subsections in Section III should be recommended.

In section IV, please justify the selection of the datasets. What features of a data set are considered valuable for video action recognition? Which were the other data sets considered? Why were the other data sets discarded?

In subsection 4.1.2, please explain what the usual sampling strategy is. Please scientifically justify the use of the AdamW optimizer. In section 4.2, what is the meaning of popular approaches? Why were the ablation studies performed with SSV2 data set?

Please insert the information regarding further works in Section V.

The references should be exhaustively reviewed. Please correct the authors' names and last names. Moreover, please complete all the references. For example, reference one only includes information regarding authors and work name. Nobody knows if it is a paper or a book. Please revise the format of the references.

In summary, the introduction did not provide sufficient background, and the research design is not clear. The methods were not adequately described, the results were unclear, and the results did not support the conclusions.

Author Response

Dear Reviewer,

Thanks a lot for the comments, which are of great help to our paper.

We are uploading our point-by-point response to the comments. Please see the attachment.

Reviewer 2 Report

This work considers the computational complexity issue in video action recognition when using the transformer. To address this problem, a new framework that uses both the spatial-windows attention and linear attention techniques has been proposed. To show the effectiveness of the proposed framework, some experimental results are provided.

The motivation of this work is valid and clear. Also, this paper technically sound. To show the effectiveness of the proposed method, extensive experiments have been implemented. However, this work can be further improved in the following aspects:

-       The organization of this work is kind of messy. It is not easy to follow. For instance, both Fig. 1 and Fig. 2 are important to illustrate the proposed framework. However, these figures are not presented in a very clear way, it would be better to reorganize these blocks and add more descriptions in the caption. Also, Sec. 3 is not stated in a clear way, which can be further improved.

-       In the experiments, four datasets have been used, and around 10 different baseline methods have been compared. However, it seems like some methods are only compared in one or two datasets, like reference [57]. Can you also provide the results of these methods on the remaining datasets?

-       In the computational complexity analyses part, there only exists the complexity calculation of the proposed method. However, why the proposed method can help to reduce the computational complexity is missing. It would be better to add some theoretical analyses (like proof or mathematical analyses) to clarify this.

-       I also have some concerns about the novelty of the proposed method. The proposed framework seems a combination of some well-developed techniques, even the structure of the framework also appears in a similar form in the existing works. Can you provide more discussion about the novelty and how the proposed framework can bring insights to the community?

-       The figures used in this work are not presented in vector graphics, it would be better to update the form of these figures.

-       The writing can be further improved, there exists some typos, also a lot of sentences have grammar issues. These should be revised in the final version.

-       The equations are supposed to be ended with either a comma or a period.

Author Response

(The authors gave the same response as above.)

Round 2

Reviewer 1 Report

The authors conducted all the suggestions offered by the reviewer. Therefore, the paper was greatly improved. There are still many typo errors to be solved and a few acronyms that were not explained.

The paper's current version is suitable for publication in the journal.

Author Response

Dear Reviewer,

Sincerely thank you for the comments, which are of great help to our paper.

We are uploading (a) our point-by-point response to the comments and (b) an updated manuscript with red font indicating changes. Please see the attachment.

Reviewer 2 Report

The authors have addressed most of my doubts. I am glad that the organization of this work has been updated and some additional computational analyses are provided. It would be better to elaborate the difference between the proposed framework and the following exsiting works:

1. Optimal lockdown for pandemic control

2. Adversarial crowdsourcing through robust rank-one matrix completion

3. Optimal vaccine allocation for pandemic stabilization

Author Response

(The authors gave the same response as above.)
